# Comparative Study of ZnO Thin Films Grown on Quartz Glass and Sapphire (001) Substrates by Means of Magnetron Sputtering and High-Temperature Annealing

**Weijia Yang [1],*, Fengming Wang [1], Zeyi Guan [1], Pengyu He [1], Zhihao Liu [1], Linshun Hu [1], Mei Chen [1], Chi Zhang [1], Xin He [1],* and Yuechun Fu [2]**

[1] School of Applied Physics and Materials, Wuyi University, Jiangmen 529020, Guangdong, China; wfm17346627784@163.com (F.W.); q145590560@163.com (Z.G.); hepengyu123@163.com (P.H.); wlxylzh607@163.com (Z.L.); hulinshun@163.com (L.H.); chenmei1106@sina.com (M.C.); chi_zhang@126.com (C.Z.)

[2] School of Resources, Environment and Materials, Guangxi University, Nanning 530004, Guangxi, China; ycfu@gxu.edu.cn

\* Correspondence: yangweijia@wyu.edu.cn (W.Y.); hexinwyu@126.com (X.H.)

**Abstract:** In this work, we reported a comparative study of ZnO thin films grown on quartz glass and sapphire (001) substrates through magnetron sputtering and high-temperature annealing. Firstly, the ZnO thin films were deposited on the quartz glass and sapphire (001) substrates in the same conditions by magnetron sputtering. Afterwards, the sputtered ZnO thin films underwent an annealing process at 600 °C for 1 h in an air atmosphere to improve the quality of the films. X-ray diffraction, scanning electron microscopy, atomic force microscopy, X-ray photoelectron spectroscopy (XPS), ultraviolet-visible spectra, photoluminescence spectra, and Raman spectra were used to investigate the structural, morphological, electrical, and optical properties of the both as-received ZnO thin films. The ZnO thin films grown on the quartz glass substrates possess a full width of half maximum value of 0.271° for the (002) plane, a surface root mean square value of 0.50 nm and O vacancies/defects of 4.40% in the total XPS O 1s peak. The comparative investigation reveals that the whole properties of the ZnO thin films grown on the quartz glass substrates are comparable to those grown on the sapphire (001) substrates. Consequently, ZnO thin films with high quality grown on the quartz glass substrates can be achieved by means of magnetron sputtering and high-temperature annealing at 600 °C.

**Keywords:** ZnO thin films; quartz glass substrate; sapphire substrate; magnetron sputtering; annealing

---

## 1. Introduction

Semiconductor materials play a critical role in modern society and have been made into light emitting diodes (LEDs) [1–3], surface acoustic wave (SAW) devices [4], sensors [5–8], solar cells [9], and so on [4,10–12], which significantly promote our life quality. ZnO is an enduring star of the third-generation semiconductor materials owing to its unique merits. First of all, ZnO is non-toxic and of perfect biocompatibility, leading to the great potential applications of biodetectors and biomedical restorative materials [13]. Furthermore, the electrical properties of ZnO can be improved by doping with other elements, and ZnO has been widely applied in LEDs [1,2], laser diodes [4], SAW devices [4], etc. ZnO is of low cost, and Al-doped ZnO is of excellent conductivity as well as cheap, which is

considered a superior substitute for the expensive indium tin oxide (ITO) transparent conductive material [14–16]. In addition, ZnO is found to be very sensitive to ultraviolet (UV) light, humidity, and gas and has been successfully applied in UV sensors [17], graphic imaging sensors [18], humidity sensors, and gas sensors [19,20]. Last but not least, it has been demonstrated that ZnO with bionic structure level holes is suitable for energy storage and utilization [21]. Therefore, ZnO materials have been favored by researchers.

In order to further utilize ZnO materials, they are always made into thin films. Of course, cost control is also considered to be a key factor for the applications of the ZnO thin films. In the past decades, ZnO thin films have been synthesized by metal-organic chemical vapor deposition (MOCVD) [22], molecular beam epitaxy (MBE) [23], pulsed laser deposition (PLD) [24], magnetron sputtering [11], thermal evaporation [25], sol–gel method [26], etc. [27,28]. MOCVD, MBE, and PLD are beneficial for forming single crystalline ZnO thin films, but the cost is usually high. Both thermal evaporation and the sol–gel method are cheap for synthesizing polycrystalline or well-preferred orientation ZnO thin films. Unfortunately, the quality of the ZnO thin films formed by these two methods needs to be further improved, and the process repeatability is still a huge challenge. On the contrary, magnetron sputtering is suitable for growing ZnO thin films with well-preferred orientation and even monocrystalline or single crystalline ZnO thin films. More importantly, the cost of the ZnO thin films grown by magnetron sputtering is relatively low, and magnetron sputtering also exhibits good process repeatability. Thus, magnetron sputtering is one of the common technologies for preparing ZnO thin films in industrial production.

It should be noted that the choice of the substrate also occupies a large chunk of the cost of ZnO thin film preparation. Both glass and sapphire are common substrates for growing ZnO thin films [17]. It is well known that the glass substrates are much cheaper than sapphire substrates. Hence, it has an important significance for growing ZnO thin films with high quality on the glass substrates. In this work, we report a comparative investigation of ZnO thin films on the quartz glass and sapphire substrates by means of magnetron sputtering and high-temperature annealing in order to optimize the technological parameters for preparing ZnO films of high quality on the quartz glass substrates.

## 2. Experimental Section

Firstly, the clean quartz glass and sapphire (001) substrates were put into the magnetron sputtering growth chamber. When the pressure of the growth chamber was pumped to $1 \times 10^{-5}$ Pa, Argon (Ar, 99.999% purity) was filled into the growth chamber to obtain a growth pressure of 0.45 Pa. Then 140 W work power was used to create Ar plasma for sputtering the ZnO (99.99% purity) ceramic target. The sputtered ZnO particles were deposited on the substrates at room temperature to form the ZnO thin films. Finally, the fabricated ZnO thin films were annealed in an air atmosphere at 600 °C for 1 h in an annealing furnace.

The as-received ZnO thin films were determined by X-ray diffraction (XRD, X'Pert Pro MFD, PANalytical, Almelo, Netherlands), scanning electron microscopy (SEM, ZEISS Sigma 500, Jena, Germany), atomic force microscopy (AFM, Agilent5500), X-ray photoelectron spectroscopy (XPS, Thermo Scientific Escalab250xi, Waltham, USA), ultraviolet-visible (UV-Vis) spectra (Shimadzu, UV2550, Kyoto, Japan), photoluminescence (PL) spectra (Edinburgh, FLS980, Edinburgh, UK), and confocal micro-Raman spectra (LabRAM HR UV-NIR, Paris, France). Before XPS detection, the as-received ZnO thin films were cut into about 1 cm × 1 cm size and washed with acetone, ethanol, and deionized water, then dried at 80 °C for 30 min in air atmosphere, and finally sputtered at several nanometers' distance from the ZnO film surface. Furthermore, before UV-Vis spectrometry, bare substrates were used to carry out baseline analyses as a reference, and then the ZnO samples were used to replace bare substrates in the test light path to obtain UV-Vis spectra of ZnO thin films.

## 3. Results and Discussion

XRD was employed to determine the structures of the as-annealed ZnO thin films grown on quartz glass and sapphire (001) substrates, as shown in Figure 1. As can be seen clearly, there were only two peaks located at 35.15° and 73.20° for the annealed ZnO thin films grown on the quartz glass substrate, which contributed to the ZnO (002) and ZnO (004) planes, respectively. Due to the quartz glass being amorphous, there was no peak ascribed to the quartz glass substrate. Similarly, ZnO (002) and ZnO (004) peaks were also found in the case of the annealed ZnO thin films grown on sapphire (001) substrate. Meanwhile, excluding the ZnO (002) and ZnO (004) peaks, only one sapphire (006) peak exists, which is accordingly assigned to the sapphire substrate. These results obviously indicate that the annealed ZnO thin films grown both on quartz glass and sapphire (001) substrates were of the preferred (002) orientation.

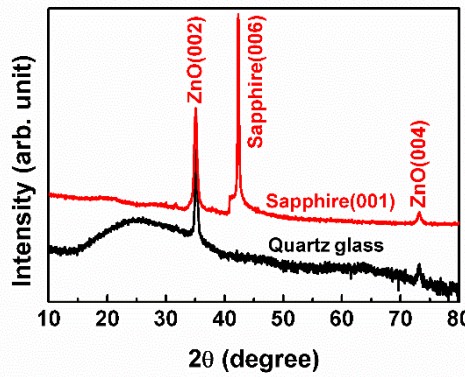

**Figure 1.** X-ray diffraction (XRD) patterns of the as-grown ZnO thin films on quartz glass and sapphire (001) substrates annealed at 600 °C for 1 h, respectively.

It should be noted that the average crystalline grain size 'D' can be extracted from the XRD results through the Scherrer's formula as follows [29]:

$$D = \frac{k\lambda}{\beta \cos \theta} \tag{1}$$

where, $k$ represents 0.9, $\lambda$ means the wavelength of CuK$_\alpha$ radiation (1.5406 Å), $\beta$ stands for the full width at half maximum (FWHM) of the (002) plane, and $\theta$ is the Bragg's diffraction angle. Additionally, the crystalline grain $D$ is believed to be associated with the amount of defects in the ZnO thin films and is considered an effective way to measure the dislocation density ($\delta$) through the following relation $\delta = 1/D^2$ [30]. Herein, $\delta$ is defined as the length of dislocation lines per unit volume of the crystal. Furthermore, the residual stress ($\sigma$) of the as-received ZnO thin films can be calculated through the following formula:

$$\sigma = -233 \left( \frac{c - c_o}{c_o} \right) \tag{2}$$

Herein, $c$ is the lattice constant of porous ZnO thin films, which is equal to $2d$ ($d$ stands for interplanar distance/spacing), while $c_o$ is the lattice constant of the perfect ZnO crystal (0.5206 nm). The calculated values of $D$, $\delta$, and $\sigma$ are listed in Table 1. Both $D$ and $\sigma$ of the as-received ZnO thin films grown on the quartz glass substrate are slightly larger than those on the sapphire (001) substrate. The FWHM and $\delta$ values of the as-received ZnO thin films grown on the quartz glass substrate are slightly smaller than those on the sapphire (001) substrate, suggesting that the ZnO thin films grown on the quartz glass substrate have better crystalline quality than those on the sapphire (001) substrate.

**Table 1.** Peak position ($2\theta$), lattice constant ($c$), full width at half maximum (FWHM), calculated grain size ($D$), dislocation density ($\delta$), and stress values ($\sigma$) of the as-received ZnO thin films grown on the quartz glass and sapphire (001) substrates with an annealing treatment of 600 °C for 1 h.

| Substrate | $2\theta/°$ | $c$(002)/nm | FWHM/° | $D$/nm | $\delta/(\times 10^{-3}$ nm) | $\sigma$/GPa |
|---|---|---|---|---|---|---|
| Quartz glass | 35.151 | 0.5132 | 0.271 | 30.8 | 1.05 | 3.312 |
| Sapphire (001) | 35.020 | 0.5137 | 0.299 | 27.9 | 1.28 | 3.088 |

SEM and AFM were employed to investigate the morphological properties of the as-received ZnO thin films grown on the quartz glass and sapphire (001) substrates. Figure 2a displays the SEM image of the ZnO thin films grown on the quartz glass substrate. As can be observed clearly, the film grown on the quartz glass is composed of ZnO nanoparticles. Besides, distinct and shallow pits turn up on the surface of the ZnO thin film. This result is comparable to the ZnO thin films annealed at 800–1000 °C by means of sol–gel method [31]. Furthermore, the cross-section SEM image indicates that the thickness of the as-received ZnO thin films on the quartz glass substrate was measured to be 102.8 nm, as shown in Figure 2c, which matches with the result (100.1 nm) detected by the profilometer. Interestingly, the surface root mean square (RMS) roughness of the ZnO thin film on the quartz glass substrate was measured to be 0.50 nm by the AFM, as illustrated in Figure 2e. On the contrary, the as-received ZnO thin film grown on the sapphire (001) substrate was also composed of smaller ZnO nano-particles but without any shallow pits on the surface (Figure 2b), suggesting a dense surface structure. As shown in Figure 2d, the thickness of the ZnO thin film on the sapphire (001) substrate was 108.6 nm, which is very closed the profilometer's result (105.3 nm). Moreover, AFM measurement revealed that the surface RMS roughness of the ZnO thin film on the sapphire (001) substrate was 0.61 nm. From what is mentioned above, it can be concluded that the as-received ZnO thin films grown on the quartz glass substrate was comparable to that on sapphire (001) substrate in terms of its morphological properties.

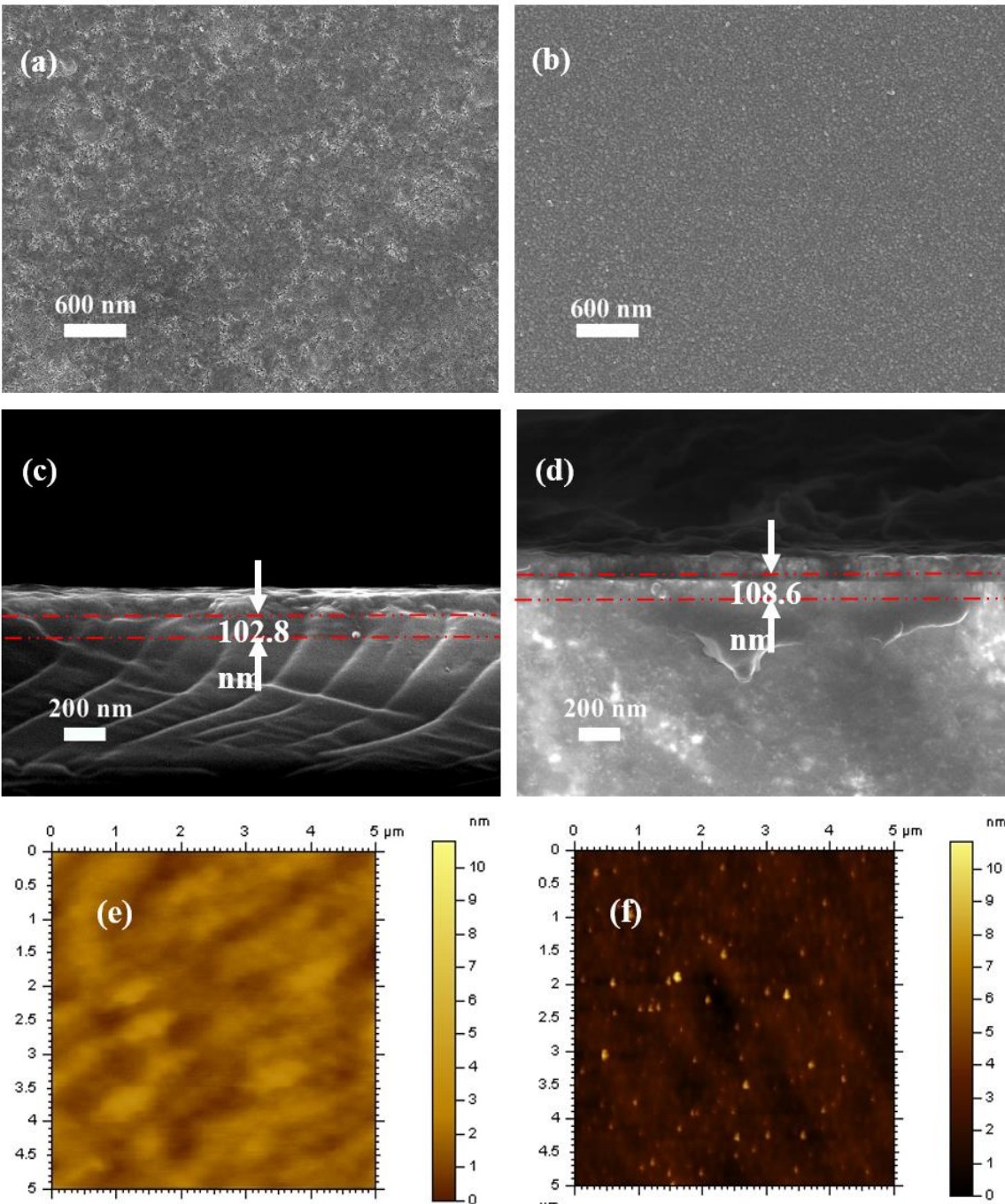

**Figure 2.** (**a**,**b**) are SEM images of the as-grown ZnO thin films on quartz glass and sapphire (001) substrates annealed at 600 °C for 1 h, respectively. (**c**,**d**) are their cross-section SEM images, respectively. (**e**,**f**) are their corresponding AFM images, respectively.

XPS has been demonstrated to be a powerful technology for confirming the chemical composition and electronic structure of the as-received ZnO thin films [32], as depicted in Figure 3a–d. Figure 3a represents the XPS analysis spectra of the as-achieved ZnO thin films grown on quartz glass and sapphire (001) substrates annealed at 600 °C for 1 h, respectively. Several peaks can be seen in Figure 3a, which correspond to the elements of Zn, C, and O, respectively. Considering that C is the standard reference, the as-received ZnO thin films are of very high purity. Furthermore, Figure 3b evaluates that the Zn $2p_{3/2}$ core levels were located in the range of 1021.24–1021.81 eV, which are assigned to the $Zn^{2+}$ ions in the ZnO thin films. As displayed in Figure 3b, two peaks located at 1021.1 and

1044.1 eV can be obviously seen, which can be attributed to the Zn 2p$_{3/2}$ and Zn 2p$_{1/2}$ electronic states, respectively. The interval between the Zn 2p$_{3/2}$ and Zn 2p$_{1/2}$ electronic states was calculated to be 23.0 eV, which was generated from the spin-orbit splitting and is the same as the previously reported value [32]. In addition, the binding energy of the Zn 2p$_{3/2}$ electronic states was slightly lower than the reported value (1021.8 eV) of the bulk ZnO [33]. XPS O 1*s* core-level spectra were performed to further understand the oxygen-related defect states in the as-received ZnO thin films, as represented in Figure 3c,d. Normally, the O 1*s* peak is composed of three Gaussian peaks: O1 (529.96 eV), O2 (530.86 eV), and O3 (531.46 eV). The low binding energy O1 peak around 529.96 eV is believed to be the response of the lattice oxygen of ZnO, Zn–O bonds. The second peak, O2, is usually located at ~530.86 eV, which is generated from the O vacancies or defects in the ZnO thin films. The third peak, O3, with a binding energy of approximately 531.46 eV is ascribed to a contribution of OH$^-$ group made from the H$_2$O in the environment. As a result, the O2 peak can be taken as an index for the O vacancies/defects in the ZnO thin films. As shown in Figure 3c and Table 2, for the ZnO thin film grown on the quartz glass substrate, the relative area ratio of the O2 peak was detected to be 4.40%, which was much smaller than that grown on the sapphire (001) substrate (33.16%). This result means that the quartz glass substrate is beneficial for reducing the O vacancies/defects in the ZnO thin films compared with the sapphire (001) substrate. It is worth noting that such a high relative area ratio value (33.16%) of the O2 peak strongly suggests a high number of internal holes in the as-received ZnO thin films grown on the sapphire (001) substrate. Therefore, the as-received ZnO thin film grown on the quartz glass substrate is much denser than that grown on the sapphire (001) substrate.

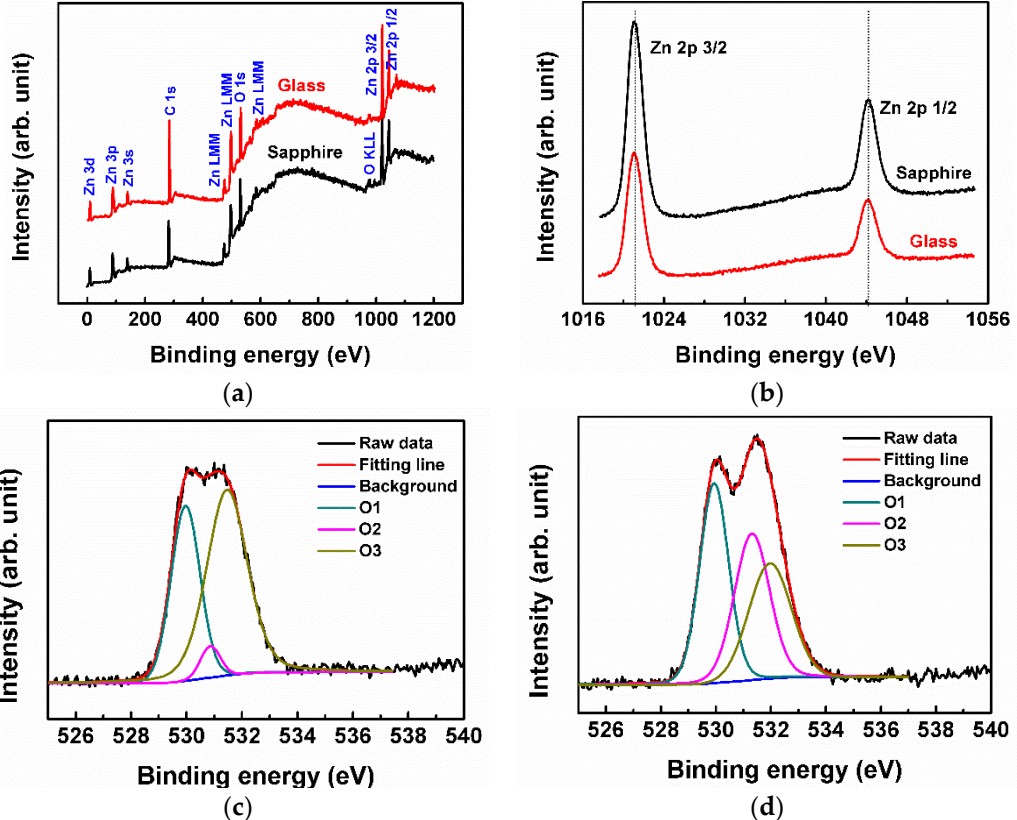

**Figure 3.** (**a**) X-ray photoelectron spectroscopy (XPS) analysis spectra and (**b**) high-resolution spectra of Zn 2*p* region for the as-achieved ZnO thin films grown on quartz glass and sapphire (001) substrates annealed at 600 °C for 1 h, respectively. (**c**) and (**d**) are XPS O 1*s* core-level spectra of the as-achieved ZnO thin films grown on quartz glass and sapphire (001) substrates annealed at 600 °C for 1 h, respectively.

**Table 2.** Oxygen-related XPS characteristic parameters for the as-grown ZnO thin films on quartz glass and sapphire (001) substrates annealed at 600 °C for 1 h.

| Substrate | Annealing | Peak | Position (eV) | FWHM (eV) | Area (%) |
|-----------|-----------|------|---------------|-----------|----------|
| Quartz glass | 600 °C | O1 | 529.96 | 1.23 | 36.32 |
| | | O2 | 530.86 | 0.85 | 4.40 |
| | | O3 | 531.46 | 1.73 | 59.28 |
| Sapphire | 600 °C | O1 | 529.94 | 1.24 | 37.06 |
| | | O2 | 531.31 | 1.51 | 33.16 |
| | | O3 | 531.97 | 1.81 | 29.78 |

UV-Vis spectra were utilized to study the optical properties of the as-received ZnO thin films grown on the quartz glass and sapphire substrates, as presented in Figure 4a,b respectively. These two ZnO thin film samples showed an optical transmittance value of approximately 83% in the wavelength range of 300–800 nm. This result is in the common level (75%–90%) of the optical transmittance for ZnO thin films [34,35]. Meanwhile, in the wavelength range of 300–370 nm, the as-received ZnO thin film on the quartz glass substrate exhibited a transmission of about 20%, which was much lower than that on the sapphire (001) substrate (approximately 60%). The higher optical transmission of the ZnO thin films on the sapphire (001) substrate may be due to the great number of internal holes of the ZnO thin films mentioned above.

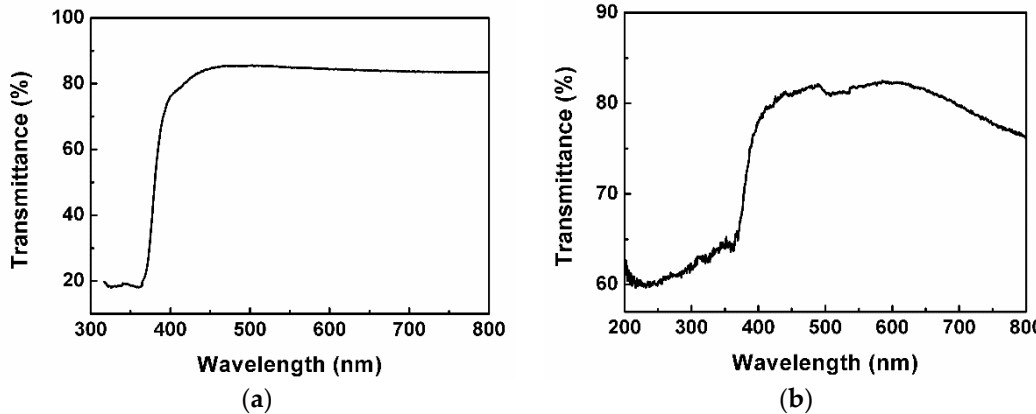

(a)  (b)

**Figure 4.** Ultraviolet-visible (UV-Vis) transmittance spectra of the as-received ZnO thin films grown on the (**a**) quartz glass and (**b**) sapphire (001) substrates, respectively.

According to a previous report, for the ZnO thin films, the optical band gap, $E_g$, can be calculated by means of the following equation [25,35]:

$$(\alpha h v)^2 = h v - E_g \tag{3}$$

where, $\alpha$ and $hv$ represent the absorption coefficient and photo energy, respectively.

As shown in Figure 5a,b, the optical band gap values of the as-received ZnO thin films grown on the quartz glass and sapphire (001) substrates were calculated to be 3.19 eV and 3.08 eV, respectively. Clearly, the optical band gap of the former was somewhat larger than that of the latter, which is attributed to the decrease in band bending effect at the grain boundaries that originate from the size increase of the ZnO nanograins [25].

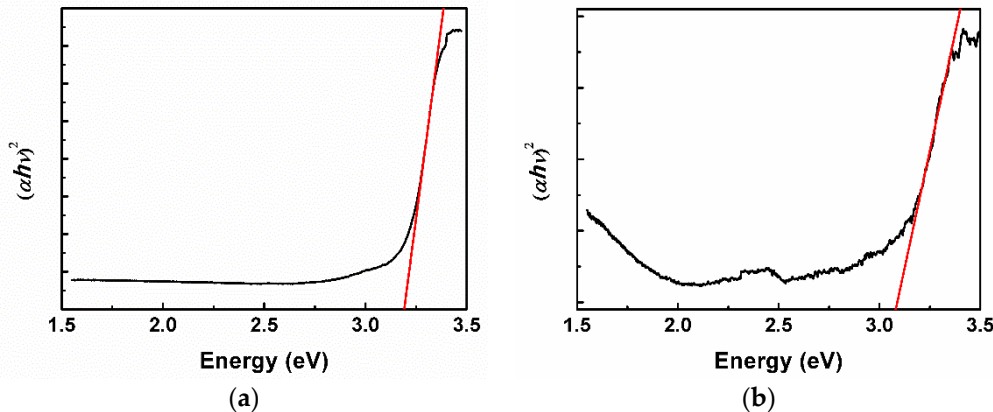

**Figure 5.** $(\alpha h\nu)^2$ vs. photon energy spectra for the ZnO thin films grown on the (**a**) quartz glass and (**b**) sapphire (001) substrates, respectively.

PL spectra were employed to investigate the emission characteristics of the as-received ZnO thin films grown on the quartz glass and sapphire (001) substrates, as illustrated in Figure 6. As can be observed clearly, both ZnO thin films revealed a strong UV peak around 350 nm, which corresponds to the near band edge emission (NBE) and zinc (Zn)-related defect luminescence peaks [36], such as Zn vacancies ($V_{Zn}$) and Zn interstitials ($Zn_i$). More clearly, the NBE is the recombination of free excitons through an exciton–exciton collision progress. There is a blue shift in the UV peak position compared with a previous report [37]. Furthermore, the UV peak intensity of the ZnO thin films on the quartz glass substrates was almost half of that on the sapphire (001) substrate. Apart from the UV peak, there was a wide and weak peak observed at 721 nm for the ZnO thin films on the quartz glass substrate, while two sharp peaks around 615 and 703 nm existed in the ZnO thin films on the sapphire (001) substrate. According to previous reports, the red peaks located at 721 or 703 nm were confirmed to be the contribution of the Zn vacancies ($V_{Zn}$) and/or complex defects involving $V_{Zn}$ [38,39]. Meanwhile, the yellow-orange peak of 615 nm was verified to correspond to oxygen interstitials ($O_i$) [40–42] and OH groups [43]. Consequently, the ZnO thin film on the quartz glass substrate mainly possessed the Zn vacancies/defects and some oxygen vacancies/defects, which is consistent with the XPS result (Figure 3c). Notably, the as-received ZnO thin film on the sapphire (001) substrate had relatively few Zn vacancies/defects and many oxygen vacancies/defects, which is well matched with the XPS measurement (Figure 3d).

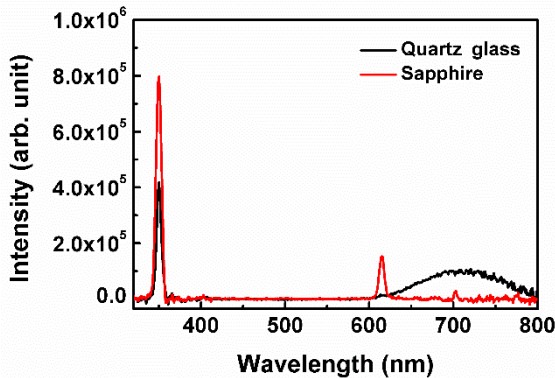

**Figure 6.** Photoluminescence (PL) spectra of the as-received ZnO thin films grown on the quartz glass and sapphire substrates.

Raman spectra were introduced to detect the residual stress in the as-received ZnO thin films grown on the quartz glass and sapphire (001) substrates at room temperature, as displayed in Figure 7. For the as-received ZnO thin film on the sapphire (001) substrate, four characteristic Raman peaks

named $E_2$(low), $A_1$(TO), $E_2$(high), and $A_1$(LO) were found, excluding the two characteristic Raman peaks (marked by points) of the sapphire substrate. In contrast, for the ZnO thin film on the quartz glass substrate, three peaks, $E_2$(low), $E_2$(high) and $A_1$(LO), were observed.

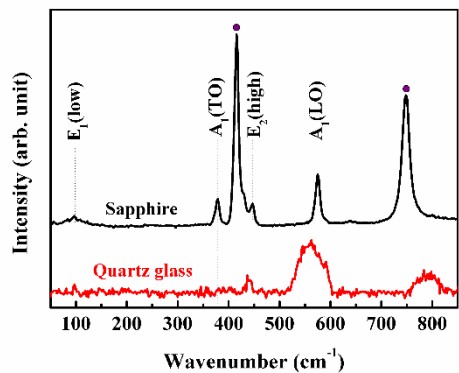

**Figure 7.** Raman spectra of the as-received ZnO thin films grown on the quartz glass and sapphire substrates, respectively.

According to previous reports, the $E_2$(high) mode located at appropriately 437 cm$^{-1}$ only corresponds to the oxygen atoms in the ZnO thin films. This clearly means that the residual stress of the wurtzite ZnO thin films can be obtained from the $E_2$(high) mode [26], owing to which it is significantly sensitive and dependent on the residual stress. More deeply, the residual stress is normally calculated from the $E_2$(high) mode based on the following equation [26]:

$$\sigma = \frac{\Delta\omega \left( \text{cm}^{-1} \right)}{4.4} \ (\text{GPa}) \tag{4}$$

where, $\sigma$ and $\Delta\omega$ are the residual stress and the shift compared with the standard $E_2$(high) mode of the stress-free ZnO bulk [4,26], respectively. The $E_2$(high) mode of the stress-free ZnO bulk is located at 437 cm$^{-1}$. In this work, the $E_2$(high) mode values of the as-received ZnO thin films on the quartz glass and sapphire (001) substrates were 441.0 cm$^{-1}$ and 447.1 cm$^{-1}$, suggesting the shifts of 4.0 cm$^{-1}$ and 10.1 cm$^{-1}$, respectively. Based on Equation (4), the residual stress values of the as-received ZnO thin films on the quartz glass and sapphire (001) substrates are 0.909 GPa and 2.295 GPa, respectively. These results are smaller than those detected by XRD. This may be attributed to the difference of the detection methods. For XRD, the scan beam is of high energy and its detection depth is over dozens of microns, which means the residual stress measured from XRD is the average residual stress of the whole film [44]. In contrast, for Raman spectra, the scan beam is low energy and its detection depth is only around the surface; hence, the residual stress extracted from the Raman spectra presents the residual stress of the film surface [45]. It is well known that the residual stress of the film surface is naturally smaller than the average residual stress of the whole film thanks to the absence of covered materials [45,46]. Possibly due to the release of the residual stress generated from annealing process as well as the different detection location [47], the residual stress (0.909 GPa) of the ZnO film surface on the quartz glass substrate was significantly smaller than that calculated from the XRD detection (3.312 GPa).

Furthermore, the $A_1$(LO) mode is normally shown in the ZnO thin film composed of nano-particles [48]. It has been demonstrated that the FWHM value of the $A_1$(LO) mode is closely associated with the degree of damage or disorder in the crystal lattice [49]. Thus, the FWHM value of this mode is considered as an index of the degree of damage or crystal disorder in the as-received ZnO thin films. As can be seen in Figure 7, the FWHM value of the $A_1$(LO) mode was about 68 cm$^{-1}$ for the ZnO thin film on the quartz glass substrate, which was much larger than that for the ZnO thin film on the sapphire (001) substrate(~15 cm$^{-1}$) but smaller than the value of a previous report (~95 cm$^{-1}$) [48].

The result evidently indicates crystal disorder in the as-received ZnO thin film on the quartz glass substrate was much poorer than that on the sapphire (001) substrate, which is in good accordance with the observations from the SEM images (Figure 2a,b).

## 4. Conclusions

We have comparatively investigated the ZnO thin films grown on the quartz glass and sapphire (001) substrates by magnetron sputtering and high-temperature annealing of 600 °C for 1 h. XRD detections revealed that the ZnO thin film grown on the quartz glass substrate had a better crystalline quality than that grown on the sapphire (001) substrate. It was found by SEM and FAM measurements that both ZnO thin films were composed of ZnO nano-particles and have a very dense as well as very smooth surface with a surface RMS roughness value of 0.50–0.65 nm. Furthermore, it was demonstrated by XPS that both ZnO thin films were of very high purity. The ZnO thin film grown on the quartz glass substrate contained much fewer O vacancies or defects than the latter, and that grown on the sapphire (001) substrate may have had a large number of internal holes. It was confirmed by UV-Vis spectra and PL spectra that the optical properties of the ZnO thin film on the quartz glass substrate was comparable to that on the sapphire (001) substrate. It is verified that the average residual stress of the ZnO thin film on the quartz glass substrate was slightly larger than that on the sapphire (001) substrate, while the surface residual stress of the former was significantly smaller than that of the latter. As a result, based on the same growth process in this work, the ZnO thin film grown on the quartz glass substrate was comparable to that grown on the sapphire (001) substrate in the morphological properties. In particular, the former was superior to the latter in the crystalline quality and the absorption of the UV light; hence, ZnO thin films with high quality could be obtained on quartz glass substrates.

**Author Contributions:** Writing-original draft, Supervision, W.Y.; Investigation, F.W.; Investigation, Z.G.; Investigation, P.H.; Investigation, Z.L.; Investigation, L.H.; Formal analysis, M.C.; Resources, C.Z.; Resources, Supervision, X.H.; Investigation, Y.F.

**Funding:** This work was funded by the Science Foundation for Young Teachers Projects of Wuyi University (2018td03); Innovation Project of Department of Education of Guangdong Province (2017KTSCX187; 2016KTSCX142; 2018KTSCX233); Innovative Leading Talents of Jiangmen (Jiangmen(2019)7); Cooperative Education Platform of Guangdong Province ([2016]31); Key Laboratory of Optoelectronic materials and Applications in Guangdong Higher Education (2017KSYS011); Science and Technology Projects of Jiangmen City (Jiangke [2016]189, [2017]307, [2017]149, [2017]268, [2018]352, [2018]359); PhD Start-up of Wuyi University (2017BS04); the College Students maker space' Innovation and Entrepreneurship Training Project of Wuyi University (18KWL01, 18KWL07); the College Students' Innovation and Entrepreneurship Training Project of Guangdong Province (201811349072); Special Fund for Scientific and Technological Innovation Cultivation of College Students in Guangdong (Special Fund for Climbing Program) (pdjh2019a0490).

**Conflicts of Interest:** The authors declare no conflict of interest.

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
