# Peer review of "Comparative Study of ZnO Thin Films Grown on Quartz Glass and Sapphire (001) Substrates by Means of Magnetron Sputtering and High-Temperature Annealing"

_applsci, doi:10.3390/app9214509_

Round 1
Reviewer 1 Report
Yang et al have performed a comparative study of ZnO thin films grown on glass and sapphire(001) substrates by mean of magnetron sputtering
and high temperature annealing. Main conclusion is that the properties of ZnO thin films grown on glass substrates are comparable to
those grown on the sapphire(001) substrates. Experimental results are sound and support the drawn conclusion.
However, I have a question
authors states "The higher optical transmission of the ZnO thin films on the sapphire(001) substrate may be owing to the great internal holes of ZnO thin films mentioned above"
what are these internal holes and where do they come from?
Besides,there are sentences throughout the manuscript that should be improved- in terms of clarity and presentation style.
Author Response
Point 1: However, I have a question authors states "The higher optical transmission of the ZnO thin films on the sapphire(001) substrate may be owing to the great internal holes of ZnO thin films mentioned above" .what are these internal holes and where do they come from?

Response 1: Thank you for your comment. These internal holes are defects in the ZnO crystal, which are formed by the large thermal and lattice mismatch stress between ZnO film and substrate as well as drastically release stress and grain rearrangement during high thermal annealing.
Point 2: Besides, there are sentences throughout the manuscript that should be improved- in terms of clarity and presentation style.
Response 2: Thank you for your suggestion. We have taken actions accordingly. The details can be found in our revised manuscript.

Reviewer 2 Report
Major comments
In the "Introduction" authors have mentioned that "we report a comparative investigation of ZnO thin films on the glass and sapphire substrates in order to seek for the optimized technological parameters of ZnO thin films with high quality on the glass substrates". Does "technological parameters" refer to the parameters of magnetron sputtering and post annealing for preparing ZnO film? The only variable parameter was the material of the substrate, and I did not notice how the technological parameters affected the structure and properties of ZnO film grown on the glass substrate.
In the "Experimental sections" authors have mentioned that "Before XPS detection, the as-received ZnO thin films were cut into about 1x1 cm2 size and washed by acetone, ethanol and deionized water, and then dried at 80 oC for 30 min". Does such a cleaning process affect the detection of elements chemical states of ZnO film surface? Why not sputter several nanometers from ZnO film surface, instead of such a cleaning process? In which kind of atmosphere the samples were dried?
In the paragraph above Figure 1, "c is the lattice constant of porous ZnO thin films, which is equal to 2d", what does "d" represent?
In the paragraph above Figure 1, "The FWHM value and δ of the former are slightly smaller than those of the latter, suggesting that the ZnO thin films grown on the glass substrate have better crystalline quality than those on the sapphire(001) substrate", how did authors obtain the value of FWHM, with peak fitting? Could the authors provide the error of estimation of FWHM? I suppose both FWHM value and δ are related to and can be replaced with "D", does crystalline quality solely depend on "D"?
In the paragraph above Figure 2, authors mentioned that "the as-received ZnO thin film grown on the sapphire(001) substrate is without any shallow pits on the surface, suggesting a dense surface structure", meanwhile, in the paragraph below Figure 2, authors have concluded that "the as-received ZnO thin film grown on the glass substrate is much dense than that grown on the sapphire(001) substrate", how would authors comment on the contradictory conclusions, driven from SEM and XPS detections? It might be necessary to adopt other methods or parameters for estimating film density, for instance, the refractive index, which can be obtained using spectroscopic ellipsometry.
In Figure 3 (d) shown the O 1s core-level spectrum of ZnO film on sapphire substrate with Gauss 3-peak fitting, how many times have authors tried to fit the spectrum? Have authors tried to do a 4-peak fitting? The spectrum visually presents sub-peaks O1, O2, O3 with similar area ratio as the spectrum of ZnO film on the glass substrate.
Why did authors choose the annealing temperature of 600 oC, which might be too high for glass? Which kind of glass have authors used?
In the present paper, authors have estimated the residual stress using Raman spectroscopy based on Raman shift of E2 mode, the shift of E2 mode only depends on the residual stress? Do other parameters, for instance, grain size, non-stoichiometry, and etc., affect the shift of E2 mode?
On the 9th page, authors have mentioned that "for the Raman spectra, the scan beam is low energy and its detected depth is only around the surface, the residual stress extracted from the Raman spectra presents the residual stress of the film surface". The detection depth of Raman spectroscopy depends on many factors, how did the author estimate or control the detection depth to "only around the surface"? Moreover, the Raman spectrum of ZnO film on a sapphire substrate shows peaks from sapphire.
Minor comments (English should be polished)
In the 1st paragraph of "Introduction", "have made into light emitting diodes, surface acoustic wave (SAW) device...", "have made into" should be replaced with "have been made into".
In the 1st paragraph of "Introduction", "ZnO owns outstanding electrical properties by doping other elements" → "electrical properties of ZnO can be improved by doping with other elements".
In the 2nd paragraph of "Introduction", "magnetron sputtering is favor to grow well preferred orientation ZnO thin films" → "magnetron sputtering is favorable to grow ZnO thin films with well preferred orientation".
In the 2nd paragraph of "Introduction", " the cost of the ZnO thin films grown by magnetron sputtering is relatively cheap", "the cost ... is cheap" → "the cost ... is low".
In the 3rd paragraph of "Introduction", "It is well-known that the glass substrates are very cheaper than sapphire substrates", "well-known" could be replaced with "well known", and "very cheaper" should be replaced with "much cheaper".
In the 3rd paragraph of "Introduction", "in order to seeking for the optimized technological parameters of ZnO thin films with high quality on the glass substrates" → "in order to optimize technological parameters for preparing ZnO films of high quality on the glass substrates".
"Experimental sections" → "Experimental section".
In the 1st paragraph of "Experimental sections", "create Ar plasmas for sputtering the ZnO (99.99% purity) ceramic target", "plasmas" → "plasma".
In the 1st paragraph of "Results and discussion", "are (002) preferred orientation" → "are with (002) preferred orientation".
In the paragraph below Scherrer formula, "whereas, k represents 0.9, λ means the wavelength of CuKα radiation (1.5406 Å), β steads for the full width at half maximum (FWHM) of the (002) plane", "whereas," → "where", "steads for" → "stands for".
In the paragraph above Figure 1, "The D, δ and σ calculated results are listed in Table 1" → " The calculated values of D, δ and σ are listed in Table 1".
"Table 1. peak position (2θ), lattice constant (c), FWHM, calculated grain size (D), dislocation density (δ) and strain values (σ) of the as-received ZnO thin films grown on the glass and sapphire(001) substrates with an annealing treatment of 600 oC for 1 h", "σ" represents stress, not strain.
In the paragraph below Table 1, "SEM and AFM are deployed to investigate the morphological properties", "deployed" → "employed".
In the paragraph below Table 1, "Furthermore, the across-section SEM image indicates that the thickness of the as-received ZnO thin films on the glass substrate is measured to be 102.8 nm", "across-section" → "cross-section".
In the paragraph below Figure 2, "Furthermore, the Zn 2p line is carried out to determine the Zn-O bonds", I do not understand what authors mean.
In the paragraph below Figure 2, "The third O3 peak with a binding energy of approximately 531.46 eV is ascribed to a contribution of OH- group made from the H2O in the environment", "made from" → "from".
In the paragraph below Figure 2, "It's worth noting that such a high relative area ratio value (33.16%) of the O2 peak", "It's" → "It is".
In the paragraph below Figure 2, " Therefore, the as-received ZnO thin film grown on the glass substrate is much dense than that grown on the sapphire(001) substrate ", "dense" → "denser".
On the 9th page, "For the XRD, the scan beam is of high energy and its detected depth is over dozens of microns, which means the residual stress measured from XRD is the average residual stress of the whole film. In contrast, for the Raman spectra, the scan beam is low energy and its detected depth is only around the surface, the residual stress extracted from the Raman spectra presents the residual stress of the film surface", "For the XRD" → "For XRD", "its detected depth" → "its detection depth", "for the Raman spectra" → "for Raman spectroscopy".

Author Response
Response to Reviewer 2 Comments
Point 1: In the "Introduction" authors have mentioned that "we report a comparative investigation of ZnO thin films on the glass and sapphire substrates in order to seek the optimized technological parameters of ZnO thin films with high quality on the glass substrates". Does "technological parameters" refer to the parameters of magnetron sputtering and post annealing for preparing ZnO film? The only variable parameter was the material of the substrate, and I did not notice how the technological parameters affected the structure and properties of ZnO film grown on the glass substrate.
Response 1: Thank you for your nice suggestion. We have taken action accordingly. We have revised the sentence to match the manuscript.
Point 2: In the "Experimental sections" authors have mentioned that "Before XPS detection, the as-received ZnO thin films were cut into about 1x1 cm2 size and washed by acetone, ethanol and deionized water, and then dried at 80 oC for 30 min". Does such a cleaning process affect the detection of elements chemical states of ZnO film surface? Why not sputter several nanometers from ZnO film surface, instead of such a cleaning process? In which kind of atmosphere the samples were dried?
Response 2: Thank you for your comment. In our work, the cleaned ZnO films were dried in air atmosphere, and finally sputtered several nanometers from ZnO film surface.
Point 3: In the paragraph above Figure 1, "c is the lattice constant of porous ZnO thin films, which is equal to 2d", what does "d" represent?
Response 3: Thank you for your comment. d stands for interplaner distance/spacing, which has added in the revised manuscript.
Point 4: In the paragraph above Figure 1, "The FWHM value and δ of the former are slightly smaller than those of the latter, suggesting that the ZnO thin films grown on the glass substrate have better crystalline quality than those on the sapphire(001) substrate", how did authors obtain the value of FWHM, with peak fitting? Could the authors provide the error of estimation of FWHM? I suppose both FWHM value and δ are related to and can be replaced with "D", does crystalline quality solely depend on "D"?
Response 4: Thank you for your nice suggestion. The FWHM values were calculated from the XRD peaks through XRD analysis software. Crystalline quality is not only consistent with "D", but also influenced with internal holes, stacking faults, dislocation density, etc.
Point 5: In the paragraph above Figure 2, authors mentioned that "the as-received ZnO thin film grown on the sapphire(001) substrate is without any shallow pits on the surface, suggesting a dense surface structure", meanwhile, in the paragraph below Figure 2, authors have concluded that "the as-received ZnO thin film grown on the glass substrate is much dense than that grown on the sapphire(001) substrate", how would authors comment on the contradictory conclusions, driven from SEM and XPS detections? It might be necessary to adopt other methods or parameters for estimating film density, for instance, the refractive index, which can be obtained using spectroscopic ellipsometry.
Response 5: Thank you for your comment. SEM observation is mainly to detect the surface morphology of ZnO thin film, while XPS results can be used to confirming the chemical composition and electronic structure of the as-received ZnO thin films, and further deduce property of whole ZnO films. It is possible that ZnO films on sapphire(001) substrates own flat surface, but have few holes in the internal. As a result, SEM results and XPS results are not contradictory. In this work, we just employ XPS results to give a simple qualitative for the density of ZnO films.
Point 6: In Figure 3 (d) shown the O 1s core-level spectrum of ZnO film on sapphire substrate with Gauss 3-peak fitting, how many times have authors tried to fit the spectrum? Have authors tried to do a 4-peak fitting? The spectrum visually presents sub-peaks O1, O2, O3 with similar area ratio as the spectrum of ZnO film on the glass substrate.
Response 6: Thank you for your nice suggestion. We have tried to fit the spectrum for about 10 times. And 4-peak fitting is also carried out. However, when the fitting is finished, the fourth peak becomes as the same as the base line. Therefore, 4-peak fitting is not suitable in this case.
Point 7: Why did authors choose the annealing temperature of 600 oC, which might be too high for glass? Which kind of glass have authors used?
Response 7: Thank you for your comment. The glass substrate in this work is quartz side glass. According to our annealing experiment, in the temperature range of 200-600 oC, crystalline quality is improved with the increase of the annealing temperature. However, when the annealing temperature is over 600 oC, the slide glass is very possible to be unstable, and even broken. Thereby, we choose the annealing temperature of 600 oC.
Point 8: In the present paper, authors have estimated the residual stress using Raman spectroscopy based on Raman shift of E2 mode, the shift of E2 mode only depends on the residual stress? Do other parameters, for instance, grain size, non-stoichiometry, and etc., affect the shift of E2 mode?
Response 8: Thank you for your nice suggestion. Based on the Raman basic principles and previous work, the shift of E2 mode only depends on the residual stress[1-3].
[1] M. Su Kim, S. Kim, J.-Y. Leem, Laser-assisted sol-gel growth and characteristics of ZnO thin films, Applied Physics Letters, 100 (2012) 252108.
[2] S. Kunj, K. Sreenivas, Residual stress and defect content in magnetron sputtered ZnO films grown on unheated glass substrates, Current Applied Physics, 16 (2016) 748-756.
[3] R. Menon, V. Gupta, H.H. Tan, K. Sreenivas, C. Jagadish, Origin of stress in radio frequency magnetron sputtered zinc oxide thin films, Journal of Applied Physics, 109 (2011) 064905.
Point 9: On the 9th page, authors have mentioned that "for the Raman spectra, the scan beam is low energy and its detected depth is only around the surface, the residual stress extracted from the Raman spectra presents the residual stress of the film surface". The detection depth of Raman spectroscopy depends on many factors, how did the author estimate or control the detection depth to "only around the surface"? Moreover, the Raman spectrum of ZnO film on a sapphire substrate shows peaks from sapphire.
Response 9: Thank you for your comment. The detection location of Raman spectra and XRD is usually not the same, which indicates that there is a certain difference in the residual stress between them[4]. Furthermore, according to previous work, the residual stress calculated from Raman spectra is usually reflected the residual stress of the film surface[4], while the residual stress calculated from XRD is the average of the whole film[5]. Meanwhile, the calculation methods of the two detection method are different, and the introduced errors are also inconsistent[5]. Therefore, the residual stress calculated from Raman spectra is different with that calculated from XRD.
[4] J.P. Tomba, M. de la Paz Miguel, C.J. Perez, Correction of optical distortions in dry depth profiling with confocal Raman microspectroscopy, Journal of Raman Spectroscopy, 42 (2011) 1330-1334.
[5] B.A. Sarsfield, M. Davidovich, S. Desikan, M. Fakes, S. Futernik, J.L. Hilden, J.S. Tan, S. Yin, G. Young, B. Vakkalagadda, K. Volk, Powder X-ray Diffraction Detection of Crystalline Phases in Amorphous Pharmceuticals, the International Centre for Diffraction Data, 2006.
Point 10: In the 1st paragraph of "Introduction", "have made into light emitting diodes, surface acoustic wave (SAW) device...", "have made into" should be replaced with "have been made into".
Response 10: Thank you for your comment. In the 1st paragraph of "Introduction", "have made into" have been replaced with "have been made into".
Point 11: In the 1st paragraph of "Introduction", "ZnO owns outstanding electrical properties by doping other elements" → "electrical properties of ZnO can be improved by doping with other elements".
Response 11: Thank you for your comment. In order to match the manuscript, we delete this part. Details can be seen in our revised manuscript.
Point 12: In the 2nd paragraph of "Introduction", "magnetron sputtering is favor to grow well preferred orientation ZnO thin films" → "magnetron sputtering is favorable to grow ZnO thin films with well preferred orientation".
Response 12: Thank you for your comment. "Magnetron sputtering is favor to grow well preferred orientation ZnO thin films" has been revised into "magnetron sputtering is favorable to grow ZnO thin films with well preferred orientation".
Point 13: In the 2nd paragraph of "Introduction", " the cost of the ZnO thin films grown by magnetron sputtering is relatively cheap", "the cost ... is cheap" → "the cost ... is low".
Response 13: Thank you for your comment. In the sentence of "the cost of the ZnO thin films grown by magnetron sputtering is relatively cheap", the “cheap” has been corrected into “low”. Details can be found in our revised manuscript.
Point 14: In the 3rd paragraph of "Introduction", "It is well-known that the glass substrates are very cheaper than sapphire substrates", "well-known" could be replaced with "well known", and "very cheaper" should be replaced with "much cheaper".
Response 14: Thank you for your comment. In the sentence of "It is well-known that the glass substrates are very cheaper than sapphire substrates", "well-known" has been replaced with "well known", and "very cheaper" has been replaced with "much cheaper". Details can be found in our revised manuscript.
Point 15: In the 3rd paragraph of "Introduction", "in order to seeking for the optimized technological parameters of ZnO thin films with high quality on the glass substrates" → "in order to optimize technological parameters for preparing ZnO films of high quality on the glass substrates".
Response 15: Thank you for your comment. In order to match the manuscript, we delete "in order to seeking for the optimized technological parameters of ZnO thin films with high quality on the glass substrates".
Point 16: "Experimental sections" → "Experimental section".
Response 16: Thank you for your comment. "Experimental sections" has been corrected into "Experimental section".
Point 17: In the 1st paragraph of "Experimental sections", "create Ar plasmas for sputtering the ZnO (99.99% purity) ceramic target", "plasmas" → "plasma".
Response 17: Thank you for your comment. In the sentence of "create Ar plasmas for sputtering the ZnO (99.99% purity) ceramic target", "plasmas" has been changed into "plasma".
Point 18: In the 1st paragraph of "Results and discussion", "are (002) preferred orientation" → "are with (002) preferred orientation".
Response 18: Thank you for your comment. In the 1st paragraph of "Results and discussion", "are (002) preferred orientation" has been changed into "are with (002) preferred orientation".
Point 19: In the paragraph below Scherrer formula, "whereas, k represents 0.9, λ means the wavelength of CuKα radiation (1.5406 Å), β steads for the full width at half maximum (FWHM) of the (002) plane", "whereas," → "where", "steads for" → "stands for".
Response 19: Thank you for your comment. In the sentence of "whereas, k represents 0.9, λ means the wavelength of CuKα radiation (1.5406 Å), β steads for the full width at half maximum (FWHM) of the (002) plane", "whereas," has been changed into "where", "steads for" has been changed into "stands for".
Point 20: I In the paragraph above Figure 1, "The D, δ and σ calculated results are listed in Table 1" → " The calculated values of D, δ and σ are listed in Table 1".
Response 20: Thank you for your comment. "The D, δ and σ calculated results are listed in Table 1" has been changed into" The calculated values of D, δ and σ are listed in Table 1".
Point 21: "Table 1. peak position (2θ), lattice constant (c), FWHM, calculated grain size (D), dislocation density (δ) and strain values (σ) of the as-received ZnO thin films grown on the glass and sapphire(001) substrates with an annealing treatment of 600 oC for 1 h", "σ" represents stress, not strain.
Response 21: Thank you for your comment. In the part of "Table 1. peak position (2θ), lattice constant (c), FWHM, calculated grain size (D), dislocation density (δ) and strain values (σ) of the as-received ZnO thin films grown on the glass and sapphire(001) substrates with an annealing treatment of 600 oC for 1 h", “strain” has been corrected into “stress”.
Point 22: In the paragraph below Table 1, "SEM and AFM are deployed to investigate the morphological properties", "deployed" → "employed".
Response 22: Thank you for your comment. In the sentence of "SEM and AFM are deployed to investigate the morphological properties", "deployed" has been changed into "employed".
Point 23: In the paragraph below Table 1, "Furthermore, the across-section SEM image indicates that the thickness of the as-received ZnO thin films on the glass substrate is measured to be 102.8 nm", "across-section" → "cross-section".
Response 23: Thank you for your comment. In the sentence of "Furthermore, the across-section SEM image indicates that the thickness of the as-received ZnO thin films on the glass substrate is measured to be 102.8 nm", "across-section" has been corrected into "cross-section".
Point 24: In the paragraph below Figure 2, "Furthermore, the Zn 2p line is carried out to determine the Zn-O bonds", I do not understand what authors mean.
Response 24: Thank you for your comment. We have revised this sentence.
Point 25: In the paragraph below Figure 2, "The third O3 peak with a binding energy of approximately 531.46 eV is ascribed to a contribution of OH- group made from the H2O in the environment", "made from" → "from".
Response 25: Thank you for your comment. In the sentence of "The third O3 peak with a binding energy of approximately 531.46 eV is ascribed to a contribution of OH- group made from the H2O in the environment", "made from" has been revised into "from".
Point 26: In the paragraph below Figure 2, "It's worth noting that such a high relative area ratio value (33.16%) of the O2 peak", "It's" → "It is".
Response 26: Thank you for your comment. In the sentence of "It's worth noting that such a high relative area ratio value (33.16%) of the O2 peak", "It's" has been changed into "It is".
Point 27: In the paragraph below Figure 2, " Therefore, the as-received ZnO thin film grown on the glass substrate is much dense than that grown on the sapphire(001) substrate ", "dense" → "denser".
Response 27: Thank you for your comment. In the sentence of " Therefore, the as-received ZnO thin film grown on the glass substrate is much dense than that grown on the sapphire(001) substrate ", "dense" has been corrected into "denser".
Point 28: On the 9th page, "For the XRD, the scan beam is of high energy and its detected depth is over dozens of microns, which means the residual stress measured from XRD is the average residual stress of the whole film. In contrast, for the Raman spectra, the scan beam is low energy and its detected depth is only around the surface, the residual stress extracted from the Raman spectra presents the residual stress of the film surface", "For the XRD" → "For XRD", "its detected depth" → "its detection depth", "for the Raman spectra" → "for Raman spectroscopy".
Response 28: Thank you for your comment. In the sentence of "For the XRD, the scan beam is of high energy and its detected depth is over dozens of microns, which means the residual stress measured from XRD is the average residual stress of the whole film. In contrast, for the Raman spectra, the scan beam is low energy and its detected depth is only around the surface, the residual stress extracted from the Raman spectra presents the residual stress of the film surface", "For the XRD" has been changed into "For XRD", "its detected depth" has revised "its detection depth", "for the Raman spectra" has been corrected into "for Raman spectroscopy".

Reviewer 3 Report
Major comments:
For the optical UV-Vis transmittance spectra, are the authors measuring just the film alone? The transmittance spectra seem to be a measurement with reference to air, meaning the transmittance is film + substrate. Using this with the optical bandgap equation, it may not be accurate. If intend to measure purely the film alone, perhaps the reference should be a bare substrate. In addition, the optical bandgap is obtained from the gradient intersecting photon energy. The gradient intersection on Figure 5(a) is reasonable giving a 3.19 eV. However, the gradient interaction of Figure 5(b) is rather “forcefully” giving a 3.14eV; a reasonable gradient will give a much smaller bandgap.Figure 4(b) has a high transmittance of ~60% at 200-400 nm which is unusual for sapphire. Do the authors have any reason on why this high transmittance is there? The ZnO films are stated to be deposited together in the same chamber and their thicknesses are also measured to be of similar thickness of ~105 ± 5 nm. Can the authors give some explanation on why is there a dip in the optical transmittance at 700nm onwards on sapphire sample only?
Glass is a very generic substrate. There are so many different kinds of glass like soda lime, borosilicate, etc. Can the authors be more specific of what kind of substrate are they using? Different glasses have various percentages of elements inside. At high temperature of 600 °C, is there any outgaussing/diffusion from the glass?
Minor comments:
There are numerous spelling typos in terms of superscript and subscript throughout the manuscript wordings and diagram captions like line 77, 83, 84,104, 108 and many more. Please check the manuscript again. The wordings of table 1 are overflowing to next line. Please check the alignment of Table 1. Since Figure 2(e) and (f) are to present the similar in their RMS value, can the authors set a similar scale bar for better comparison?Author Response
Point 1: For the optical UV-Vis transmittance spectra, are the authors measuring just the film alone? The transmittance spectra seem to be a measurement with reference to air, meaning the transmittance is film + substrate. Using this with the optical bandgap equation, it may not be accurate. If intend to measure purely the film alone, perhaps the reference should be a bare substrate. In addition, the optical bandgap is obtained from the gradient intersecting photon energy. The gradient intersection on Figure 5(a) is reasonable giving a 3.19 eV. However, the gradient interaction of Figure 5(b) is rather “forcefully” giving a 3.14eV; a reasonable gradient will give a much smaller bandgap.
Response 1: Thank you for your nice suggestion. In our work, the transmittance spectra of ZnO thin films by taking the reference of bare substrates. Furthermore, we have corrected the fitting line and revised Figure 5(b), and the bandgap of sapphire sample is corrected to be 3.08 eV.
Point 2: Figure 4(b) has a high transmittance of ~60% at 200-400 nm which is unusual for sapphire. Do the authors have any reason on why this high transmittance is there? The ZnO films are stated to be deposited together in the same chamber and their thicknesses are also measured to be of similar thickness of ~105 ± 5 nm. Can the authors give some explanation on why is there a dip in the optical transmittance at 700nm onwards on sapphire sample only?
Response 2: Thank you for your comment. The high transmittance at 200-400 nm maybe due to in the internal of ZnO films on sapphire(001) substrates. A dip in the optical transmittance at 700nm onwards on sapphire sample only maybe due to the microstructure (nanoparticles) on the surface of ZnO film on sapphire substrate, as shown in Figure 2(f). According to our previous work, owing to the mirostruture on the surface, the transmittance is decreased as the increase of wavelength at the range over 700 nm. As a comparison, there is no obvious microstructure on the surface of ZnO films on glass substrates, as shown in Figure 2(e).
Point 3: Glass is a very generic substrate. There are so many different kinds of glass like soda lime, borosilicate, etc. Can the authors be more specific of what kind of substrate are they using? Different glasses have various percentages of elements inside. At high temperature of 600 °C, is there any outgaussing/diffusion from the glass?
Response 3: Thank you for your nice suggestion. The glass substrate in this work is quartz side glass. Based on our previous work, quartz side glass is stable after 600 oC annealing. According to our annealing experiment, in the range from 200 to 600 oC, crystalline quality is increased with the increase of the annealing temperature. However, when the annealing temperature is over 600 oC, the slide glass is very possible to be unstable, and even broken. Thereby, we choose the annealing temperature of 600 oC. We are very sorry, it is difficult for us to carried out the test of outgaussing/diffusion from the glass at 600 oC.
Point 4: here are numerous spelling typos in terms of superscript and subscript throughout the manuscript wordings and diagram captions like line 77, 83, 84,104, 108 and many more. Please check the manuscript again. The wordings of table 1 are overflowing to next line. Please check the alignment of Table 1. Since Figure 2(e) and (f) are to present the similar in their RMS value, can the authors set a similar scale bar for better comparison?
Response 4: Thank you for your comment. We have check the manuscript and fixed spelling errors. The details can be found in our revised manuscript. Furthermore, we have revised the scale bar in Figure 2(e) to match that in Figure 2(f). The details can be found in our revised manuscript.

Round 2
Reviewer 2 Report
The authors substantially rewrote the article, but a few minor comments remained.
Comment on Response 4.
As shown in Your manuscript, δ solely depends on the value of D, and the FWHM, I suppose, also corresponds to D, so check the logic of the sentence "The FWHM value and δ of the former are slightly smaller than those of the latter, suggesting that the ZnO thin films grown on the glass substrate have better crystalline quality than those on the sapphire(001) substrate" and Your response to my question. Rewrite the sentence.
Comment on Response 7.
Replace "glass" with "quartz glass" in Your manuscript.
Comment on Response 9.
There is no the statement that "for the Raman spectra, the scan beam is low energy and its detected depth is only around the surface, the residual stress extracted from the Raman spectra presents the residual stress of the film surface" in the article "Correction of optical distortions in dry depth profiling with confocal Raman microspectroscopy". Perhaps I missed it, so would you mind pointing it out or explaining how you have driven this conclusion?
Have you conducted confocal Raman microspectroscopy as described in "Correction of optical distortions in dry depth profiling with confocal Raman microspectroscopy"? If so, please point it out in the Experimental section.
The thickness of your films is much less than the values mentioned in "Correction of optical distortions in dry depth profiling with confocal Raman microspectroscopy", you have controlled the detection depth to dozens of nanometers or even several nm?
Additional comment.
At the end of the second paragraph of the Experimental section, "Furthermore, before UV-vis spectra,", and then what?

Author Response
Point 1: As shown in Your manuscript, δ solely depends on the value of D, and the FWHM, I suppose, also corresponds to D, so check the logic of the sentence "The FWHM value and δ of the former are slightly smaller than those of the latter, suggesting that the ZnO thin films grown on the glass substrate have better crystalline quality than those on the sapphire(001) substrate" and Your response to my question. Rewrite the sentence.
Response 1: Thank you for your nice suggestion. Both δ and FWHM are can be employed to indicate the crystalline quality of ZnO thin films. The smaller values of δ and FWHM are, the better the crystalline quality of ZnO thin films is. However, there is no direct relationship between FWHM and D. To make it clearly, we have corrected the sentence in our revised manuscript. The details are as followed. “The FWHM value and δ of the as-received ZnO thin films grown on the quartz glass substrate are slightly smaller than those on the sapphire(001) substrate, suggesting that the ZnO thin films grown on the quartz glass substrate have better crystalline quality than those on the sapphire(001) substrate.”
Point 2: Replace "glass" with "quartz glass" in Your manuscript.
Response 2: Thank you for your nice suggestion. We have replaced "glass" with "quartz glass" in our revised manuscript.
Point 3: There is no the statement that "for the Raman spectra, the scan beam is low energy and its detected depth is only around the surface, the residual stress extracted from the Raman spectra presents the residual stress of the film surface" in the article "Correction of optical distortions in dry depth profiling with confocal Raman microspectroscopy". Perhaps I missed it, so would you mind pointing it out or explaining how you have driven this conclusion?
Response 3: Thank you for your comment. In order to explain the residual stress difference clearly, we have revised these sentences and replaced the references with other new suitable references. The details are as followed. “In contrast, for Raman spectra, the scan beam is low energy and its detection depth is only around the surface, the residual stress extracted from the Raman spectra presents the residual stress of the film surface [45]. It is well known that the residual stress of the film surface is naturally smaller than the average residual stress of the whole film thanks to the absence of covered materials [45, 46]. Especially, possibly due to the release of the residual stress generated from annealing process as well as the different detection location [47], the residual stress (0.909 GPa) of the ZnO film surface on the quartz glass substrate is significantly smaller than that (3.312 GPa) calculated from the XRD detection.”
[45] T. Wermelinger, F.C.F. Mornaghini, C. Hinderling, R. Spolenak, Correlation between the defect structure and the residual stress distribution in ZnO visualized by TEM and Raman microscopy, Materials Letters, 64 (2010) 28-30.
[46] P. Jannotti, G. Subhash, J. Zheng, V. Halls, Measurement of microscale residual stresses in multi-phase ceramic composites using Raman spectroscopy, Acta Materialia, 129 (2017) 482-491.
[47] Z. Wang, B. Si, S. Chen, B. Jiao, X. Yan, A nondestructive raman spectra stress 2D analysis for the pressure sensor sensitive silicon membrane, Engineering Failure Analysis 105 (2019) 1252-1261.
Point 4: Have you conducted confocal Raman microspectroscopy as described in "Correction of optical distortions in dry depth profiling with confocal Raman microspectroscopy"? If so, please point it out in the Experimental section.
Response 4: Thank you for your nice suggestion. We have taken action accordingly. In this work, confocal Micro-Raman spectra were used to test the ZnO thin films. Details can be found in our revised manuscript: “The as-received ZnO thin films were determined by X-ray diffraction (XRD, X’Pert Pro MFD), scanning electron microscopy (SEM, ZEISS Sigma 500), atomic force microscopy (AFM, Agilent5500), X-ray photoelectron spectroscopy (XPS, Thermo Scientific Escalab250xi), Ultraviolet-visible (UV-Vis) spectra (Shimadzu, UV2550), Photoluminescence (PL) spectra (Edinburgh, FLS980), and confocal Micro-Raman spectra (LabRAM HR UV-NIR). ”
Point 5: The thickness of your films is much less than the values mentioned in "Correction of optical distortions in dry depth profiling with confocal Raman microspectroscopy", you have controlled the detection depth to dozens of nanometers or even several nm?
Response 5: Thank you for your comment. In this work, confocal Micro-Raman spectra were used to test the ZnO thin films. Based on our experiment and previous work [45, 46], the result calculated from Raman spectra is the residual stress of the surface. However, I am not very sure the exact detection depth of Raman spectra.
[45] T. Wermelinger, F.C.F. Mornaghini, C. Hinderling, R. Spolenak, Correlation between the defect structure and the residual stress distribution in ZnO visualized by TEM and Raman microscopy, Materials Letters, 64 (2010) 28-30.
[46] P. Jannotti, G. Subhash, J. Zheng, V. Halls, Measurement of microscale residual stresses in multi-phase ceramic composites using Raman spectroscopy, Acta Materialia, 129 (2017) 482-491.
Point 6: At the end of the second paragraph of the Experimental section, "Furthermore, before UV-vis spectra,", and then what?
Response 6: Thank you for your nice suggestion. We have corrected the sentence in our manuscript. The details are as followed. “Furthermore, before UV-Vis spectra, bare substrates were used to carry out baseline as a reference, and then the ZnO samples were used to replace bare substrates in test light path to obtain UV-Vis spectra of ZnO thin films.”
